# RIS-Assisted D2D Communication over Nakagami-*m* Fading with RSMA

**DOI:** 10.3390/s24113423

**Published:** 2024-05-26

**Authors:** Yunhao Ding, Linfei Chen, Peishun Yan, Wei Duan

**Affiliations:** 1School of Information Science and Technology, Nantong University, Nantong 226019, Chinasinder@ntu.edu.cn (W.D.); 2School of Electrical Engineering and Automation, Engineering Training Center, Nantong University, Nantong 226019, China; 3Educational and Scientific Institute of Energy Saving and Energy Management, National Technical University of Ukraine “Igor Sikorsky Kyiv Polytechnic Institute”, 03056 Kyiv, Ukraine

**Keywords:** reconfigurable intelligent surface (RIS), rate-splitting multiple access (RSMA), Nakagami-*m* fading channels, achievable ergodic rate, maximum ratio combination (MRC)

## Abstract

In this study, we investigated reconfigurable intelligent surface (RIS)-assisted device-to-device (D2D) communication systems over Nakagami-*m* fading channels. To enhance the reliability of RIS-assisted D2D communications, we utilized the rate-splitting multiple access (RSMA) technique to maximize the achievable ergodic rate for our considered systems. Specifically, both devices decoded the common symbol by treating private symbols as interference, and then each private symbol was decoded by treating the other as interference. In order to maximize the achievable ergodic rate at the destination, we analyzed the achievable ergodic rate of the RIS link and the D2D link, and the destination jointly decoded both symbols transmitted from the source and device by involving the maximum ratio combination (MRC). We obtained a closed-form expression for the achievable ergodic rate of the proposed RIS-assisted D2D communication system. Finally, we investigated the influence of power allocation factors and the number of reflective elements on the achievable ergodic rate. As seen by the numerical results, there was a good match between the analysis and simulation results, as well as significant superiority compared with existing works.

## 1. Introduction

In recent years, device-to-device (D2D) communication has been widely regarded as one of the key technologies in 5G. However, D2D communication is largely limited by the propagation environment. As an emerging technology, a reconfigurable intelligent surface (RIS) is able to dynamically and intelligently regulate the channel environment, which can be used to solve the issue of a poor propagation environment [1]. An RIS is considered a significant breakthrough in existing wireless communication technologies [2], particularly playing a crucial role in the 5G and upcoming 6G eras [3]. An RIS consists of a large number of low-cost passive reflective elements integrated on a plane, which can be controlled through a pre-programmed controller [4]. Each reflective component of an RIS can independently adjust the amplitude and/or phase of the incident signal so that it achieves precise three-dimensional reflection beamforming. Moreover, it is able to intelligently reconfigure the wireless propagation environment, which makes the channel state information (CSI) controllable and significantly improves the performance of wireless communication networks [5]. In [6], the closed-form expressions of the outage probability and channel capacity over Rice fading channels are provided. The authors derived a closed-form expression for the tight upper bound of the achievable ergodic rate of RIS-assisted D2D systems over Nakagami-*m* fading channels [7]. In [8], an efficient active elements selection (AES) algorithm was proposed to enhance the system performance for hybrid RIS-assisted D2D communication systems, where a subset of RIS elements was selected from the whole elements set to connect with power amplifiers, serving as the active RIS, while the others reflected the incident signals passively, serving as the passive RIS. In [9], the achievable ergodic rate of cooperative relay systems over Nakagami-*m* fading channels were analyzed. A novel RIS-aided joint simultaneous wireless information and power transfer (SWIPT)–mobile edge computing (MEC) system was considered in [10], where the user could maximize the saved energy by optimizing the RIS reflection coefficients to simultaneously cater to both energy harvesting and task offloading. Ref. [11] proposes a novel approximate method to evaluate the channel performance of cascaded RIS-aided wireless networks with phase errors over Nakagami-*m* fading channels. Novel RIS-based energy harvesting (EH) systems with linear EH (L-EH) and non-linear EH (NL-EH) models over Nakagami-*m* fading channels are presented in [12], where the closed-form expressions of throughput, outage probability, and average harvested power were derived.

With the exponential growth of device connectivity, data traffic will also result in much interference [13]. Therefore, a technology is needed to improve anti-interference performance while simultaneously maintaining high spectral efficiency and high energy efficiency during large-scale connections [14]. Non-orthogonal multiple access (NOMA) provides a higher data rate, spectral efficiency, and energy efficiency compared with traditional orthogonal multiple access (OMA), which can be used to solve the aforementioned problem [15]. However, NOMA has a serious problem, which requires a large number of successive interference cancellation (SIC) operations, which will result in high computational complexity. For a large number of users, some users in NOMA must decode their own information under high interference from other users. To overcome this problem, rate-splitting multiple access (RSMA) technology was proposed. RSMA adopts a single-layer SIC, where common user information is extracted and divided into separate common information flows, and each user’s private information is divided into multiple private information flows [16]. Compared with NOMA, the interference in RSMA is reduced by partially treating the interference as noise [17,18]. Multiple studies showed that RSMA is a strong contender for the future of wireless communications. For example, ref. [19] solved the non-convex sum-rate maximization problem in RSMA first. The authors showed that the uplink RSMA is more reliable compared with uplink NOMA in communication with finite block length codes [20]. Ref. [21] investigated RSMA in short-packet communication and demonstrated that RSMA with smaller block lengths achieves the same maximum–minimum fairness rate as NOMA and space division multiple access (SDMA). Ref. [22] first adopted the data-oriented approach in downlink RSMA systems. The authors addressed the problem of the orthogonal frequency division multiplexing (OFDM) waveform under linear time-varying (LTV) channels by considering RSMA in [23]. Although RSMA is an important multiple access technology that outperforms other contemporary multiple access techniques in terms of throughput, it still requires low latency and high reliability.

In [24], an RIS-assisted unmanned aerial vehicle (UAV)-based vehicular communication system was investigated, and the outage performance of the availability of direct UAV-desired vehicle equipment (DVE) link under an interference-limited scenario was analyzed. The RIS-assisted RSMA communication system where a base station broadcast signals to multiple users with a dedicated RIS was considered in [25]. The expression of the outage probability was derived for the scenarios of optimal and discrete phase shifts. Referring to [24,25], they mainly investigated the outage performance of RIS-assisted communication systems with RSMA, while we utilized the maximum ratio combination (MRC) to obtain a higher achievable ergodic rate and derived the closed-form expression of it.

According to the above studies, an RIS is clearly an effective technology that can be used to improve the communication environment of D2D systems. In this study, we attempted to analyze the achievable ergodic rate of the proposed RIS-assisted D2D communication system over a Nakagami-*m* fading channel, which can model general communication environments, including Rayleigh and Rician fading. The main contributions are summarized as follows:We propose an RIS-assisted communication system consisting of a source, an RIS, and two near and far devices, where the far device is beyond the coverage of the source, and the RIS is used to assist the far device communication. The achievable ergodic rate of the proposed system over a Nakagami-*m* fading channel was analyzed.We carried out the analysis of the achievable ergodic rate of RIS and D2D links with RSMA and then combined their signals by an MRC scheme, which could help to obtain a higher achievable ergodic rate compared with conventional RIS-assisted D2D communication systems. Furthermore, we derived the closed-form expression of the achievable ergodic rate at the destination, where our proposed scheme performed better than the previous benchmark.The influence of power allocation factors and the number of reflective elements on the achievable ergodic rate were also investigated, where the optimal number of RIS elements was found for the proposed system. The numerical results verified the correctness of our analysis, as well as the superiority of the proposed scheme compared with a conventional scheme.

## 2. System Model

As shown in Figure 1, we considered a cooperative scenario that consists of a source *S*, an RIS *R*, and two devices $D_{1}$ and $D_{2}$. *S* and both devices $D_{1}$ and $D_{2}$ are equipped with a single antenna. The RIS containing *N* passive reflective elements is deployed between *S* and $D_{2}$, and it is closer to *S*. In addition, $D_{1}$ is another device between *S* and $D_{2}$, where it receives a signal from *S* and transmits the signal to $D_{2}$. Since the channel *S* to $D_{2}$ is blocked, $D_{2}$ is beyond the coverage of *S*. In a conventional D2D communication system, the destination-received signal is only from device $D_{1}$. This means that the achievable ergodic rate at the destination is low. Thus, the transmission from *S* to $D_{2}$ requires the assistance of an RIS to help improve the performance of the wireless communication. The channels *S*–*R*, *R*–$D_{2}$, *S*–$D_{1}$, and $D_{1}$–$D_{2}$ are respectively denoted by $h_{SR}\in C^{N\times 1}$, $h_{RD_{2}}\in C^{N\times 1}$, $h_{{SD}_{1}}$, and $h_{D_{1}D_{2}}$, which follow independent and identical Nakagami-*m* distributions.

There are two phases involved in the whole transmission. In the first phase, the source broadcasts symbols $s_{c}$, $s_{1}$, and $s_{2}$ to the RIS and $D_{1}$, where $s_{c}$ stands for the common symbol and $s_{1}$ and $s_{2}$ are private symbols for $D_{1}$ and $D_{2}$, respectively. By adopting the superposition coding, the transmitted signal is in the from of $\sqrt{a_{c}*P_{t}}s_{c}+\sqrt{a_{1}*P_{t}}s_{1}+\sqrt{a_{2}*P_{t}}s_{2}$, where $a_{c}$, $a_{1}$, and $a_{2}$, with $a_{c}$ + $a_{1}$ + $a_{2}$ = 1, denote the power allocation factors, and $P_{t}$ stands for the total transmitted power. Since the decoding scheme is RSMA, $s_{c}$ should be allocated more transmission power than $s_{1}$ and $s_{2}$, i.e., $a_{c}$ > $a_{1}$ + $a_{2}$. The received signal at $D_{1}$ is thus given by
(1)$$\begin{array}{cc} y_{SD_{1}} & =h_{SD_{1}}\left(\sqrt{a_{c}*P_{t}}s_{c}+\sqrt{a_{1}*P_{t}}s_{1}+\sqrt{a_{2}*P_{t}}s_{2}\right)+n_{SD_{1}}, \end{array}$$
where $n_{\delta }$, for δ$\in \{SR,SD_{1},RD_{2},D_{1}D_{2}\}$, is the additive white Gaussian noise (AWGN), which has a zero mean and variance $\sigma^{2}$. Then, the RIS reflects the symbols to $D_{2}$ in the same phase, where the signals at $D_{2}$ reflected from the RIS can be given by
(2)$$\begin{array}{cc} \; & y_{RD_{2}}=h_{RD_{2}}^{H}\Theta h_{SR}\left(\sqrt{a_{c}*P_{t}}s_{c}+\sqrt{a_{1}*P_{t}}s_{1}+\sqrt{a_{2}*P_{t}}s_{2}\right)+n_{SR}, \end{array}$$
the reflecting coefficients can be fully represented by the diagonal matrix $\Theta =\eta \mathrm{diag}(\exp^{j\theta_{1}},\cdots ,\exp^{j\theta_{N}})$, where η∈(0,1] is the fixed amplitude reflection coefficient and $\left\{\theta_{1},\cdots ,\theta_{N}\right\}$ are the phase-shift variables that can be optimized by the RIS.

In the second phase, $D_{1}$ forwards the decoded symbols $s_{c}$ and $s_{2}$ to $D_{2}$. The transmitted signal is in the form of $\sqrt{a_{c}*P_{t}}$$s_{c}$ + $\sqrt{\left(1-a_{c}\right)*P_{t}}$$s_{2}$. The signals at $D_{2}$ transmitted from $D_{1}$ are given by
(3)$$\begin{array}{cc} \; & y_{D_{1}D_{2}}=h_{D_{1}D_{2}}\left(\sqrt{a_{c}*P_{t}}s_{c}+\sqrt{\left(1-a_{c}\right)*P_{t}}s_{2}\right)+n_{D_{1}D_{2}}. \end{array}$$

### 2.1. Recent Scheme Revisited

$D_{2}$ decodes $s_{c}$ and $s_{2}$ from the RIS and $D_{1}$ decodes $s_{c}$, $s_{1}$, and $s_{2}$ from the source in the first phase. Since the decoding scheme is RSMA, $D_{2}$ decodes the common symbol $s_{c}$ first by treating $s_{1}$ and $s_{2}$ as noise and then decodes the private symbol $s_{2}$ by treating $s_{1}$ as noise, where the signal-to-interference-plus-noise ratios (SINRs) for $s_{c}$ and $s_{2}$ at $D_{2}$ are respectively given by
(4)$$\begin{array}{cc} \; & \gamma_{D_{2}}^{C(I)}=\frac{P_{t}a_{c}{\left|h_{RD_{2}}^{H}\Theta h_{SR}\right|}^{2}}{P_{t}a_{1}{\left|h_{RD_{2}}^{H}\Theta h_{SR}\right|}^{2}+P_{t}a_{2}{\left|h_{RD_{2}}^{H}\Theta h_{SR}\right|}^{2}+\sigma^{2}}, \end{array}$$
(5)$$\begin{array}{cc} \; & \gamma_{D_{2}}^{P(I)}=\frac{P_{t}a_{2}{\left|h_{RD_{2}}^{H}\Theta h_{SR}\right|}^{2}}{P_{t}a_{1}{\left|h_{RD_{2}}^{H}\Theta h_{SR}\right|}^{2}+\sigma^{2}}. \end{array}$$
By the same scheme, the SINRs for $s_{c}$, $s_{1}$, and $s_{2}$ at $D_{1}$ can be
(6)$$\begin{array}{cc} \; & \gamma_{D_{1}}^{C}=\frac{P_{t}a_{c}{\left|h_{SD_{1}}\right|}^{2}}{P_{t}a_{1}{\left|h_{SD_{1}}\right|}^{2}+P_{t}a_{2}{\left|h_{SD_{1}}\right|}^{2}+\sigma^{2}}, \end{array}$$
(7)$$\begin{array}{cc} \; & \gamma_{D_{1}}^{P_{1}}=\frac{P_{t}a_{1}{\left|h_{SD_{1}}\right|}^{2}}{P_{t}a_{2}{\left|h_{SD_{1}}\right|}^{2}+\sigma^{2}}, \end{array}$$
(8)$$\begin{array}{cc} \; & \gamma_{D_{1}}^{P_{2}}=\frac{P_{t}a_{2}{\left|h_{SD_{1}}\right|}^{2}}{P_{t}a_{1}{\left|h_{SD_{1}}\right|}^{2}+\sigma^{2}}. \end{array}$$
The received SINRs of $s_{c}$ and $s_{2}$ at $D_{2}$ in the second phase are respectively given by
(9)$$\begin{array}{cc} \; & \gamma_{D_{2}}^{C(II)}=\frac{P_{t}a_{c}{\left|h_{D_{1}D_{2}}\right|}^{2}}{P_{t}\left(1-a_{c}\right){\left|h_{D_{1}D_{2}}\right|}^{2}+\sigma^{2}}, \end{array}$$
(10)$$\begin{array}{cc} \; & \gamma_{D_{2}}^{P(II)}=\frac{P_{t}\left(1-a_{c}\right){\left|h_{D_{1}D_{2}}\right|}^{2}}{\sigma^{2}}. \end{array}$$
To ensure the decoding correctness, the achievable SNRs for $s_{c}$ and $s_{2}$ at the D2D link should be
(11)$$\begin{array}{cc} \; & \gamma_{C}^{rec}=\min\left(\gamma_{D_{1}}^{C},\gamma_{D_{2}}^{C(II)}\right), \end{array}$$
(12)$$\begin{array}{cc} \; & \gamma_{P}^{rec}=\min\left(\gamma_{D_{1}}^{P_{2}},\gamma_{D_{2}}^{P(II)}\right). \end{array}$$

### 2.2. Proposed Receiver Design

In the RIS-assisted D2D system, the channel of *S*–$D_{2}$ is blocked, for which we need an RIS and another device to help transmit the symbols. To enhance the reliability, we leveraged RSMA to decode the received signals. In particular, the common symbol $s_{c}$ is decoded first at either $D_{1}$ or $D_{2}$, and the private symbol follows after the SIC. The MRC is adopted by combining $y_{RD_{2}}$ and $y_{D_{1}D_{2}}$ to develop spatial diversity and improve the transmission SINRs of the symbols $s_{c}$ and $s_{2}$ at $D_{2}$.

The achievable rate of $s_{c}$ is given by $R_{c}=\frac{1}{2}\log_{2}\left(1+\gamma_{c}\right)$, where
(13)$$\begin{array}{cc} \; & \gamma_{c}=\gamma_{D_{2}}^{C(I)}+\gamma_{C}^{rec}, \end{array}$$
and that of $s_{2}$ is $R_{2}=\frac{1}{2}\log_{2}\left(1+\gamma_{2}\right)$, where
(14)$$\begin{array}{cc} \; & \gamma_{2}=\gamma_{D_{2}}^{P(I)}+\gamma_{P}^{rec}. \end{array}$$

## 3. Achievable Ergodic Rate Analysis

In this section, we analyze the achievable ergodic rate of the RIS and D2D channels, respectively. To simplify the analysis, the ideal passive beamforming (IPB) with perfect channel estimation (PCE) is considered at the RIS, and all elements have the same reflection amplitude. Note that the equivalent channel *S*–*R*–$D_{2}$ is $|h_{RD_{2}}^{H}\Theta h_{SR}|=\eta \sum_{n=1}^{N}e^{j\theta_{n}}{\left[h_{RD_{2}}\right]}_{n}{\left[h_{SR}\right]}_{n}$. Since the functions $\log_{2}\left(1+x\right)$, $\frac{a_{c}x}{(a_{1}+a_{2})x+1}$, and $\frac{a_{2}x}{a_{1}x+1}$ are all monotonically increasing functions, their related composite functions (Equations (4) and (5)) are also monotonically increasing. Consequently, the maximum rate will be achieved when the phase shifts are selected as $\theta_{n}^{*}=arg\left(h_{SD_{2}}\right)-arg\left({\left[h_{RD_{2}}\right]}_{n},{\left[h_{SR}\right]}_{n}\right)$, which means that every term has the same phase as $h_{SD_{2}}$.

Therefore, we can obtain Equations (15) and (16), which are given by
(15)$$\begin{array}{cc} \gamma_{D_{2}}^{C(I)} & =\max_{\theta_{1},\ldots ,\theta_{N}}\frac{P_{t}a_{c}{\left|h_{RD_{2}}^{H}\Theta h_{SR}\right|}^{2}}{P_{t}a_{1}{\left|h_{RD_{2}}^{H}\Theta h_{SR}\right|}^{2}+P_{t}a_{2}{\left|h_{RD_{2}}^{H}\Theta h_{SR}\right|}^{2}+\sigma^{2}}\\ \; & =\frac{P_{t}a_{c}(\eta \sum_{n=1}^{N}|{\left[h_{RD_{2}}\right]}_{n}{\left[h_{SR}\right]}_{n}{\left|\right)}^{2}}{P_{t}a_{1}(\eta \sum_{n=1}^{N}|{\left[h_{RD_{2}}\right]}_{n}{\left[h_{SR}\right]}_{n}{\left|\right)}^{2}+P_{t}a_{2}(\eta \sum_{n=1}^{N}|{\left[h_{RD_{2}}\right]}_{n}{\left[h_{SR}\right]}_{n}{\left|\right)}^{2}+\sigma^{2}}. \end{array}$$
(16)$$\begin{array}{cc} \gamma_{D_{2}}^{P(I)} & =\max_{\theta_{1},\ldots ,\theta_{N}}\frac{P_{t}a_{2}{\left|h_{RD_{2}}^{H}\Theta h_{SR}\right|}^{2}}{P_{t}a_{1}{\left|h_{RD_{2}}^{H}\Theta h_{SR}\right|}^{2}+\sigma^{2}}\\ \; & =\frac{P_{t}a_{2}(\eta \sum_{n=1}^{N}|{\left[h_{RD_{2}}\right]}_{n}{\left[h_{SR}\right]}_{n}{\left|\right)}^{2}}{P_{t}a_{1}(\eta \sum_{n=1}^{N}|{\left[h_{RD_{2}}\right]}_{n}{\left[h_{SR}\right]}_{n}{\left|\right)}^{2}+\sigma^{2}}. \end{array}$$

### 3.1. Analysis for RIS

For an RIS-assisted D2D communication system over Nakagami-*m* fading, we first denote $Y=\eta |h_{SR}{|}_{n}{\left|h_{RD_{2}}\right|}_{n}$ and $X=\sum_{n=1}^{N}Y$. Since the channels are independent, we obtain
(17)$$\begin{array}{cc} \; & E\left[Y\right]=\eta E[|h_{SR}{|}_{n}]E[|h_{RD_{2}}{|}_{n}], \end{array}$$
(18)$$\begin{array}{cc} \; & D\left[Y\right]=\eta^{2}E[|h_{SR}{|}_{n}^{2}]E[|h_{RD_{2}}{|}_{n}^{2}]-(\eta E[|h_{SR}{|}_{n}]E[|h_{RD_{2}}{|}_{n}{])}^{2}. \end{array}$$
The Nakagami-*m* fading channel can be expressed by
(19)$$\begin{array}{cc} \; & f_{|h_{i}{|}^{2}}\left(h\right)={\left(\frac{m_{i}}{\Omega_{i}}\right)}^{m_{i}}\frac{h^{m_{i}-1}}{\Gamma (m_{i})}exp\left(-\frac{m_{i}}{\Omega_{i}}h\right), \end{array}$$
where $m_{i}$ indicates the fading severity parameter of the link and $\Omega_{i}$ stands for the scale parameter for $i\in \{h_{SR},h_{RD_{2}},h_{{\mathrm{SD}}_{1}},h_{D_{1}D_{2}}\}$. Through mathematical deduction, we can can further obtain
(20)$$\begin{array}{cc} \; & E[|h_{i}\left|\right]=\frac{\Gamma (m_{i}+\frac{1}{2})}{\Gamma (m_{i})}{\left(\frac{\Omega_{i}}{m_{i}}\right)}^{\frac{1}{2}}, \end{array}$$
(21)$$\begin{array}{cc} \; & E[|h_{i}{|}^{2}]=\Omega_{i}. \end{array}$$
Denoting $\mu_{X}=\sum_{n=1}^{N}E\left[Y\right]$ and $\delta_{X}^{2}=\sum_{n=1}^{N}D\left[Y\right]$ as the mean and variance of X, from [7], we can obtain the expressions of $\mu_{X}$ and $\delta_{X}^{2}$ as
(22)$$\begin{array}{cc} \; & \mu_{X}=\eta \sum_{n=1}^{N}\frac{\Gamma \left(m_{SR,n}+\frac{1}{2}\right)\Gamma \left(m_{RD_{2},n}+\frac{1}{2}\right)}{\Gamma \left(m_{SR,n}\right)\Gamma \left(m_{RD_{2},n}\right)}{\left(\frac{\Omega_{SR,n}\Omega_{RD_{2},n}}{m_{SR,n}m_{RD_{2},n}}\right)}^{\frac{1}{2}}, \end{array}$$
and
(23)$$\begin{array}{cc} \; & \delta_{X}^{2}=\eta^{2}\sum_{n=1}^{N}\Omega_{SR,n}\Omega_{RD_{2},n}\left(1-\frac{\Gamma^{2}\left(m_{SR,n}+\frac{1}{2}\right)\Gamma^{2}\left(m_{RD_{2},n}+\frac{1}{2}\right)}{m_{SR,n}m_{RD_{2},n}\Gamma^{2}\left(m_{SR,n}\right)\Gamma^{2}\left(m_{RD_{2},n}\right)}\right). \end{array}$$
Using Jensen’s inequality:(24)$$\begin{array}{cc} \; & E\left[log_{2}\left(1+\omega \right)\right]\leq log_{2}\left(1+E\left[\omega \right]\right), \end{array}$$
the achievable ergodic rate at the RIS channel can be expressed by
(25)$$\begin{array}{cc} \; & r^{*}=\frac{1}{2}log_{2}\left(1+E\left[\gamma^{*}\right]\right), \end{array}$$
where $r^{*}\in \left\{r_{c}^{RIS},r_{p}^{RIS}\right\}$ and $\gamma^{*}\in \left\{\gamma_{D_{2}}^{C(I)},\gamma_{D_{2}}^{P(I)}\right\}$. Since $\frac{a_{1}x}{(a_{2}+a_{3})x+1}$ and $\frac{a_{3}x}{a_{2}x+1}$ are concave, Jensen’s inequality is used to provide the upper bounds on their expected values:(26)$$\begin{array}{cc} E[\gamma_{D_{2}}^{C(I)}] & =E[\frac{X^{2}a_{c}P_{t}}{X^{2}\left(a_{1}+a_{2}\right)P_{t}+\sigma^{2}}]\\ \; & \leq \frac{E\left[X^{2}\right]a_{c}P_{t}}{E\left[X^{2}\right]\left(a_{1}+a_{2}\right)P_{t}+\sigma^{2}}, \end{array}$$
and
(27)$$\begin{array}{cc} E[\gamma_{D_{2}}^{P(I)}] & =E[\frac{X^{2}a_{2}P_{t}}{X^{2}a_{1}P_{t}+\sigma^{2}}]\\ \; & \leq \frac{E\left[X^{2}\right]a_{2}P_{t}}{E\left[X^{2}\right]a_{1}P_{t}+\sigma^{2}}. \end{array}$$
According to
(28)$$\begin{array}{cc} \; & E\left[X^{2}\right]=\mu_{X}^{2}+\delta_{X}^{2}=\eta^{2}\sum_{n=1}^{N}\Omega_{SR}\Omega_{RD_{2}}, \end{array}$$
the expected values of $\gamma_{D_{2}}^{C(I)}$ and $\gamma_{D_{2}}^{P(I)}$ can be expressed as
(29)$$\begin{array}{cc} \; & E\left[\gamma_{D_{2}}^{C(I)}\right]\leq \frac{\eta^{2}\sum_{n=1}^{N}\Omega_{SR}\Omega_{RD_{2}}a_{c}P_{t}}{\eta^{2}\sum_{n=1}^{N}\Omega_{SR}\Omega_{RD_{2}}\left(a_{1}+a_{2}\right)P_{t}}, \end{array}$$
(30)$$\begin{array}{cc} \; & E\left[\gamma_{D_{2}}^{P(I)}\right]\leq \frac{\eta^{2}\sum_{n=1}^{N}\Omega_{SR}\Omega_{RD_{2}}a_{2}P_{t}}{\eta^{2}\sum_{n=1}^{N}\Omega_{SR}\Omega_{RD_{2}}a_{1}P_{t}}. \end{array}$$
From Equations (26) and (27), it is clear that $\frac{a_{1}x}{(a_{2}+a_{3})x+1}$ and $\frac{a_{3}x}{a_{2}x+1}$ monotonically increase; therefore, we further derive its achievable ergodic rate, which is shown as
(31)$$\begin{array}{cc} R_{RIS} & =r_{c}^{RIS}+r_{p}^{RIS}\\ \; & =\frac{1}{2}\log_{2}\left(1+E\left[\gamma_{D_{2}}^{C(I)}\right]\right)+\frac{1}{2}\log_{2}\left(1+E\left[\gamma_{D_{2}}^{P(I)}\right]\right)\\ \; & =\frac{1}{2}\log_{2}\left(1+\frac{\eta^{2}\sum_{n=1}^{N}\Omega_{SR}\Omega_{RD_{2}}a_{c}P_{t}}{\eta^{2}\sum_{n=1}^{N}\Omega_{SR}\Omega_{RD_{2}}\left(a_{1}+a_{2}\right)P_{t}}\right)\\ \; & \;+\frac{1}{2}\log_{2}\left(1+\frac{\eta^{2}\sum_{n=1}^{N}\Omega_{SR}\Omega_{RD_{2}}a_{2}P_{t}}{\eta^{2}\sum_{n=1}^{N}\Omega_{SR}\Omega_{RD_{2}}a_{1}P_{t}}\right). \end{array}$$

### 3.2. Analysis for D2D Communication

In this section, the achievable ergodic rate of the D2D communication is characterized over Nakagami-*m* fading channels. For better readability, we donate $\lambda_{i}=\rho \beta_{i}$, where $\beta_{i}={\left|h_{i}\right|}^{2}$. Since $\beta_{i}∼G\left(m_{i},\Omega_{i}\right)$ holds, $\lambda_{i}∼G\left(m_{i},\rho \Omega_{i}\right)$, and the probability distribution function (PDF) and cumulative distribution function (CDF) of $\lambda_{i}$ can respectively be given as
(32)$$\begin{array}{cc} f(\lambda_{i}) & ={\left(\frac{m_{i}}{\rho \Omega_{i}}\right)}^{m_{i}}\frac{\lambda_{i}^{m_{i}-1}}{\Gamma (m_{i})}\exp\left(-\frac{\lambda_{i}m_{i}}{\rho \Omega_{i}}\right),\lambda_{i}\geq 0, \end{array}$$
(33)$$\begin{array}{cc} F(\lambda_{i}) & =1-\exp\left(-\frac{\lambda_{i}m_{i}}{\rho \Omega_{i}}\right)\sum_{k=0}^{m_{i}-1}\frac{{\left(\lambda_{i}m_{i}\right)}^{k}}{{\left(\rho \Omega_{i}\right)}^{k}k!},\lambda_{i}\geq 0, \end{array}$$
where $\rho =\frac{P_{t}}{\delta^{2}}$ stands for the transmission SNR. The achievable ergodic rate of the D2D links can be rewritten as
(34)$$\begin{array}{cc} R_{D2D} & =\frac{1}{2}\log\left(1+\gamma_{C}^{rec}\right)+\frac{1}{2}\log\left(1+\gamma_{P}^{rec}\right)\\ \; & =\frac{1}{2}\underset{r_{c}}{\underset{︸}{\log\left(1+min\left(\frac{a_{c}\lambda_{SD_{1}}}{\left(1-a_{c}\right)\lambda_{SD_{1}}+1},\frac{a_{c}\lambda_{D_{1}D_{2}}}{\left(1-a_{c}\right)\lambda_{D_{1}D_{2}}+1}\right)\right)}}\\ \; & \;+\frac{1}{2}\underset{r_{p}}{\underset{︸}{\log\left(1+min\left(\frac{a_{2}\lambda_{SD_{1}}}{a_{1}\lambda_{SD_{1}}+1},\left(1-a_{c}\right)\lambda_{D_{1}D_{2}}\right)\right)}}. \end{array}$$

The closed form of the achievable ergodic rate of the D2D link can be given as
(35)$$\begin{array}{cc} \; & R_{D2D}=\frac{1}{2}\min\left\{r_{c}^{case1},r_{c}^{case2}\right\}+\frac{1}{2}\min\left\{r_{p}^{case1},r_{p}^{case2}\right\}, \end{array}$$
where $r_{c}^{case1},r_{c}^{case2},r_{p}^{case1}$, and $r_{p}^{case2}$ can be obtained by Equations (A6)–(A9), which are found in Appendix A. Similarly, the achievable ergodic rate of the RIS-assisted D2D communication system can be calculated by
(36)$$\begin{array}{cc} \; & R_{sum}=\frac{1}{2}\min\left\{r_{c}^{case1},r_{c}^{case2}\right\}+r_{c}^{RIS}+\frac{1}{2}\min\left\{r_{p}^{case1},r_{p}^{case2}\right\}+r_{p}^{RIS}, \end{array}$$
where $r_{c}^{RIS}$ + $r_{p}^{RIS}$ can be obtained by Equation (Equation 31).

## 4. Numerical Results

In this section, we provide the results of simulations performed to evaluate the performance of the considered RIS-assisted D2D communication system over Nakagami-*m* fading. In addition, *m* was set as 2 for all links; Ω was set as 1 for the *S*–*R* link and 5 for *R*–$D_{2}$, *S*–$D_{1}$, and $D_{1}$–$D_{2}$ links; $\delta^{2}$ was set to −70 dBm; the reflection coefficient $\eta_{i}$ was set to 1; and the power allocation coefficients were set to $a_{c}$ = 0.7, $a_{1}$ = 0.1, and $a_{2}$ = 0.2. All the Monte Carlo results were obtained over $10^{6}$ independent realizations. Note that all the parameters, such as the channel fading parameter and reflective coefficient, as well as the network topologies, could be set arbitrarily.

Figure 2 displays the analytical achievable ergodic rate under different numbers of reflective elements, which were set to $\left(N_{1},N_{2},N_{3}\right)=\left(100,200,300\right)$. The curves of the Monte Carlo simulations were also plotted to evaluate the correctness of the closed-form expressions. The simulations and analysis given in the graph legend respectively indicate the Monte Carlo simulation results and the results of our analytical closed form of the achievable ergodic rate, which are presented in Equation (Equation 36). In the high-SNR region, the analytical achievable ergodic rate was generally close to the Monte Carlo simulations under different numbers of reflective elements, demonstrating the correctness of our theoretical analysis. Figure 3 compares our proposed RIS-assisted D2D communication system to a conventional D2D communication system, where the number of reflective elements was set to *N* = 100, showing that our proposed scheme was significantly superior to existing works. It is obvious that an RIS is an excellent technology that can be used to effectively improve the communication performance of D2D systems.

Figure 4 plots the achievable ergodic rate in devices $D_{1}$ and $D_{2}$ under different private power allocation factors. We set the power allocation of the common symbol $s_{c}$ to $a_{c}=0.7$. The figure illustrates that as the achievable ergodic rate of $D_{2}$ increased, the rate of $D_{1}$ decreased since given the power allocation factor of common symbol was fixed, the power allocation factors of the two private symbols assumed the inverse correlation. Thus, the impact of the achievable ergodic rate in $D_{1}$ should be considered while improving the rate of $s_{1}$ based on fairness.

The relationship between the achievable ergodic rate and the number of reflective elements is demonstrated with different SNRs in Figure 5, assuming that (SNR_1_, SNR_2_, SNR_3_) = (5, 15, 25). It is clearly observed that the achievable ergodic rate increased rapidly when the number *N* belonged to 50–70, and increased a little in other intervals. Therefore, we worked out the optimal number of RIS reflective elements in this system to be N=70, which could effectively improve the achievable ergodic rate while reduce the waste of the RIS reflective elements.

## 5. Conclusions

In this study, we focused on RIS-assisted D2D communication systems over Nakagami-*m* fading. We derived the closed-form expression of the achievable ergodic rate for an RIS link and a D2D link. We used simulation results to verify the numerical solution and investigated the influence of power allocation factors on the achievable ergodic rate. In the high-SNR region, the achievable rate became a constant value. Furthermore, we compared our proposed RIS-assisted D2D scheme to a conventional D2D scheme, which showed significant superiority compared with existing works. Furthermore, we also studied the relationship between the achievable ergodic rate and the number of reflective elements, where the optimal number of RIS elements was found for the proposed system. Finally, in future works, we will analyze RIS-assisted and D2D systems in near-field communication and the system performance will be analyzed.

## Figures and Tables

**Figure 1 sensors-24-03423-f001:**
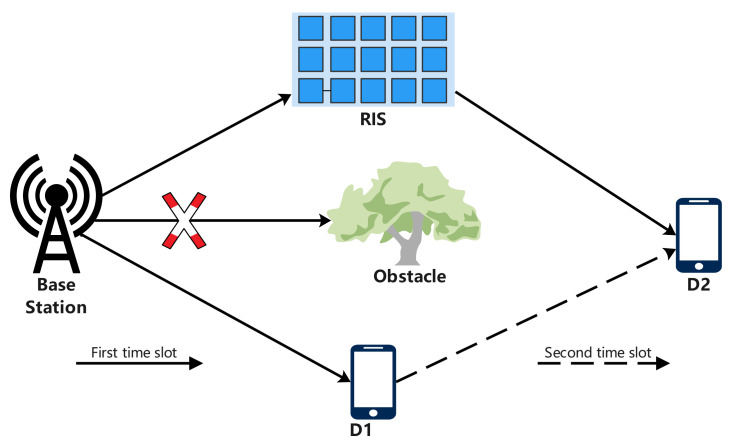
RIS-assisted D2D communication system.

**Figure 2 sensors-24-03423-f002:**
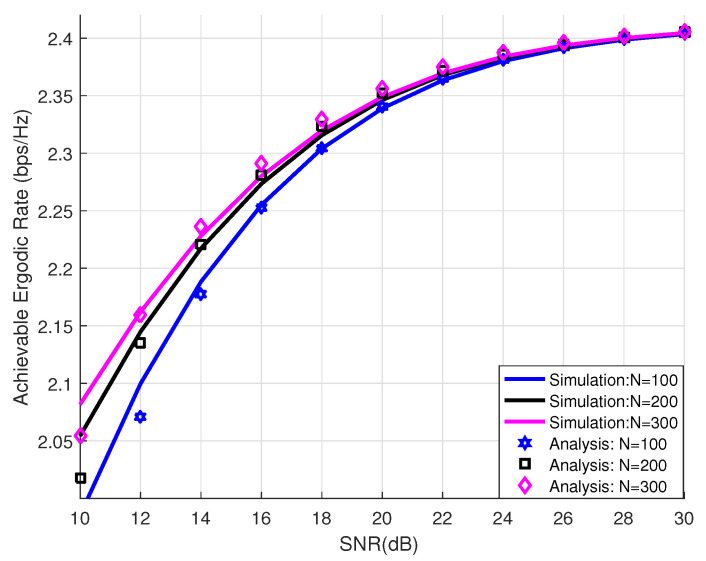
Analysis and simulation results for our proposed scheme versus SNR with different *N* values.

**Figure 3 sensors-24-03423-f003:**
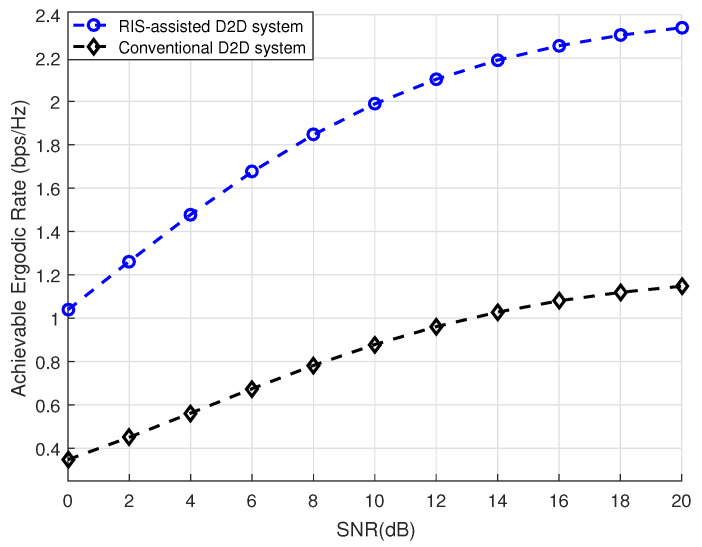
Proposed RIS-assisted D2D system and conventional D2D system versus SNR.

**Figure 4 sensors-24-03423-f004:**
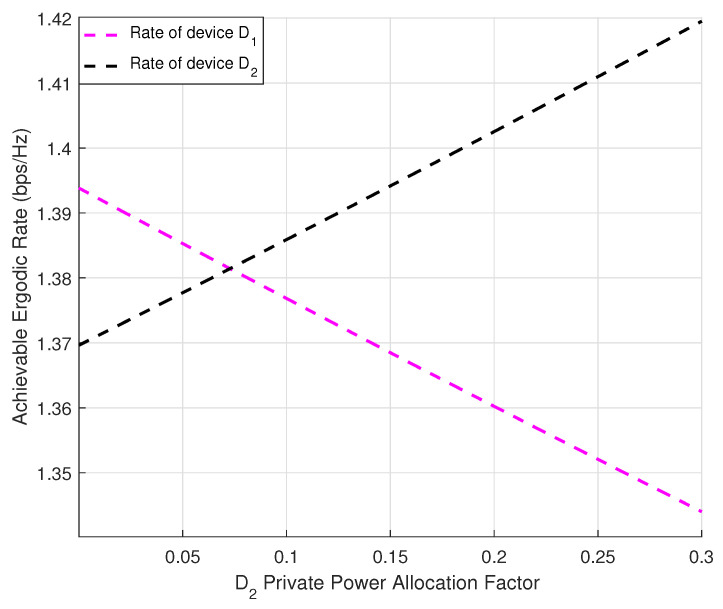
Achievable ergodic rates of $D_{1}$ and $D_{2}$ versus $a_{2}$.

**Figure 5 sensors-24-03423-f005:**
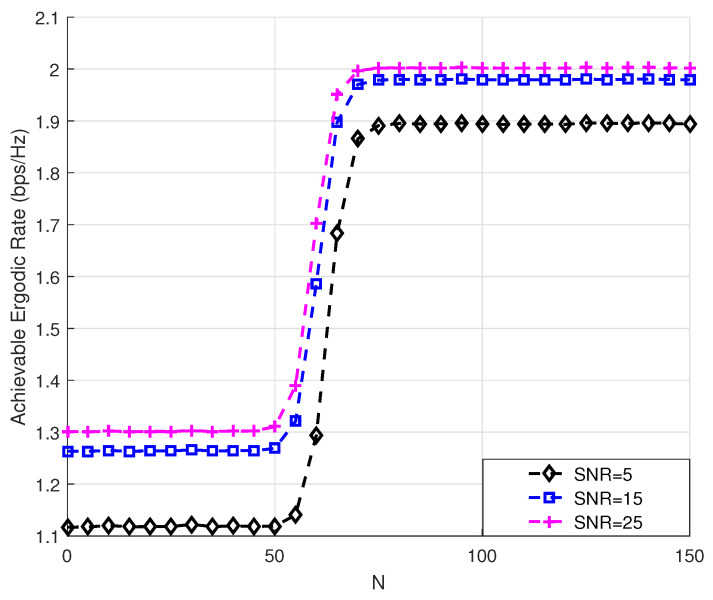
Achievable ergodic rate versus the number of RIS elements with different SNRs.

## Data Availability

Data are contained within the article.

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
