# Peer review of "RIS-Assisted D2D Communication over Nakagami-m Fading with RSMA"

_sensors, 2024, doi:10.3390/s24113423_

Round 1

Reviewer 1 Report

Comments and Suggestions for Authors Strengths:

I enjoyed reviewing your paper. Thank you for submitting. About this research: The research integrates Reconfigurable Intelligent Surface with device-to-device communications to enhance reliability in challenging environments, marking a significant advancement for 5G and 6G networks. It provides a thorough analysis with solid mathematical bases for system performance evaluation through closed-form expressions for the ergodic rate over Nakagami-m fading channels. Employing Rate Splitting Multiple Access reduces the computational complexity found in technologies like NOMA, streamlining the decoding process for systems with many users. Extensive simulations validate the theoretical models, demonstrating the system’s advantages through detailed results and graphical representations. The findings offer practical insights for enhancing D2D communications with actionable recommendations for network design, including power allocation and reflective elements optimization.

Weaknesses and Areas to improve:

The research assumes ideal conditions such as perfect channel estimation, which might not hold in real-world scenarios. Exploring the effects of non-ideal conditions could enhance future studies. The complexity of the mathematical analysis might limit its accessibility to practitioners with less mathematical expertise. Potential hardware challenges and scalability issues related to RIS deployment in actual network environments are overlooked. More comprehensive comparisons with other recent technologies could better position the advantages and limitations of the proposed system. Additionally, incorporating analyses of the system’s energy consumption could align with the growing emphasis on sustainable technology solutions, making future research more robust and applicable, potentially leading to practical implementations of RIS-assisted D2D communication systems.

Reviewer 2 Report

Comments and Suggestions for Authors

The paper presents a novel approach for enhancing reliability in Reconfigurable Intelligent Surface (RIS) assisted Device-to-Device (D2D) communication systems over Nakagami-m fading channels using Rate Splitting Multiple Access (RSMA) technique. The proposed method is interesting and addresses a pertinent issue in wireless communication. The paper is well-written, with clear organization. Additionally, it provides a solid mathematical formulation to support the proposed approach. Overall, this paper might make contributions to the field, but it presents some issues that need to be addressed before it can be considered for publication.

1) In my view there is a lack of clarity in articulating the contribution of the paper with the related literature. The paper appears to address an important topic in the wireless communication field, but the authors fail to explicitly state their contribution. Additionally, the absence of a comprehensive literature review section makes it challenging to evaluate the novelty and relevance of the study within the field.

2) Despite the authors have clearly stated the scenario within which they are operating in the article, it is not evident to this reviewer the true impact of studying this scenario in the literature. A more thorough contextualization would be necessary.

3) perhaps a typo, "frist phase" appears several times all over the document.

4) "In the high SNR region, the analytical achievable ergodic rate presented in (46) is generally close to … " 

What do you mean by (46) in this sentence? Is it a mention for equation 46? Maybe you should rephrase it to clarify.  

5) I recommend the authors give more textual attention to the introduction of the plots shown in the paper as well as more textual explanations on  principal outcomes the reader can get from the plots. The text regarding the results graphs are too raw.

6) An example (but not limited to) of my last comment: what does "Analysis" mean in the graph legend? Please consider enriching the graph legends and figures captions with more meaningful texts.

Reviewer 3 Report

Comments and Suggestions for Authors

In this paper, reconfigurable intelligent surface assisted device-to-device systems over Nakagami-m fading channels are analyzed.

The paper is generally well written, and the topic is worth of investigation. The mathematical analysis seems to be correct, and numerical results are presented in the intuitive form. However, some mistakes have to be corrected before the publication.

Although the numerical results are the more interesting part of the paper (e.g. figure 5 is very interesting), I have some doubt about the results in Figures 2 and 3. In the description, it was written that both figures corresponds to RIS-assisted D2D systems over Nakagami-m fading (and in Figure 3 the comparison with conventional D2D system was also given). However, when N=100 and 10dB<SNR<20dB, the values on the y-axis are not same in blue line in Figure 2 and blue dashed line in Figure 3. Can you explain this?

There are a lot of syntax errors in the paper, and I suggest the authors to use a spell checker to find and correct them. In the abstract, the word "intererence" is used instead of "interference". In the Section 1, "fristly" is used instead of "firstly". In Section 2, "frist" is used instead of "first",  "shuold" instead of "should", "anothor" instead of "another". In the conclusion, "shceme" is used instead of "scheme".

Furthermore, some expressions are not clear. In Eq. (20), the PDF for h_i is given. In other expressions, counter "i" never appears. Maybe it is better to define PDF for h_n? In the summation over n in Eqs. (23) and (24), it is not clear how the terms depend on counter n.

Technical preparation also has to be improved. In Section 2, page 2, line 2, in variable "D2" number 2 should be placed in the subscript. All variables should be written in italic format (e.g. variable "S"). There is no need to place space before comma (last sentence in Section 1), and space have to be placed after comma (sentence before Eq. (15)). All variables has to be defined (e.g. vector h_S_R, written with boldface letters was not formally defined). There is an extra dot at the end of the first line in Eq. (16). There is an extra space before the dot at the end of Eq. (20). In figures 2 and 3, in the x-label @db@ should be replaced with "dB".

The list of references was not written according to the MDPI template. In some references, names of the journals was written in the shorter form, in the other the full name was used. Generally, the unified approach should be accepted.

Comments on the Quality of English Language

There are a lot of syntax errors in the paper, and I suggest the authors to use a spell checker to find and correct them. In the abstract, the word "intererence" is used instead of "interference". In the Section 1, "fristly" is used instead of "firstly". In Section 2, "frist" is used instead of "first",  "shuold" instead of "should", "anothor" instead of "another". In the conclusion, "shceme" is used instead of "scheme".

Reviewer 4 Report

Comments and Suggestions for Authors

The topic of the submission is quite timely and interesting in view of RIS-assisted communications.

The overall impression from the presented submission is quite satisfactory. The text in general is well-written and well-formatted (but there are some minor issues in this part). The authors' logic and all the derivations are generally clear. The obtained results are somewhat interesting.

Although, there are some issues with the submission that prevent me from recommending it for publication without proper modifications.

Major issues:

1. In the introduction section, the authors had stated that “According to the above researches, most studies dedicated to investigate RIS assisted D2D systems over Rayleigh channel fading and Rice channel fading.”. This statement is totally wrong, since a simple search in IEEEXplore gives about 100 papers that study different aspects of RIS assuming the Nakagami model.

Well, maybe the authors kept in mind specifically RSMA, then the number of papers shrink to several dozens (still, quite a large number). So, a profound comparison with the existing results must be presented. For example, how does the current submission differ from:

 [1]  Bansal, N. Agrawal and K. Singh, "Rate-Splitting Multiple Access for UAV-Based RIS-Enabled Interference-Limited Vehicular Communication System," in IEEE Transactions on Intelligent Vehicles, vol. 8, no. 1, pp. 936-948, Jan. 2023, doi: 10.1109/TIV.2022.3168159.

[2]  D. Shambharkar, S. Dhok and P. K. Sharma, "Performance Analysis of RIS Assisted RSMA Communication System," 2022 National Conference on Communications (NCC), Mumbai, India, 2022, pp. 227-232, doi: 10.1109/NCC55593.2022.9806718.

2. After careful reading, the problem under consideration looks exactly as it is formulated for RIS/Relay communications, but here we have another device acting as a relay. In such an aspect, the problem is not new at all. There is a plethora of research dealing with it. So, the authors must clearly state what is the novelty of their problem (and hence results) compared to the classical RIS/Relay problem.

3. RSMA algorithms are quite promising for future communications, but with some serious issues (that are not addressed in the paper at all). For example, high computational costs; this ruins the main concepts of the submission, since the decoding should be done on a mobile device acting as a relay. This drawback must be carefully addressed.

4. In view of the above, the contribution of the submission seems to be negligible compared to the existing results. Anyway, it must be clearly stated at the end of the introduction section.

Minor issues:

5. Math derivations are quite tedious but fairly straightforward, thus can be easily moved to the appendices.

6.  Please, avoid using “*” as a multiplication sign.

7. Avoid using math notations (not yet introduced) in the abstract.

8. The overall writing must be improved. A lot of grammar errors are present.

9. Please, on the plot use conventional units "dB", not some strange "db".

Comments on the Quality of English Language

A lot of grammar errors are present.

Round 2

Reviewer 2 Report

Comments and Suggestions for Authors

The author have addressed my concerns.

Reviewer 3 Report

Comments and Suggestions for Authors

The authors have addressed all of my comments, the paper can be accepted in the current form.

Reviewer 4 Report

Comments and Suggestions for Authors

The authors carefully addressed my concerns, I see no other problems with publishing the submission.